# Evaluation of the Immunosafety of Cucurbit[n]uril In Vivo

**DOI:** 10.3390/pharmaceutics16010127

**Published:** 2024-01-19

**Authors:** Ekaterina Pashkina, Alina Aktanova, Olga Boeva, Maria Bykova, Elena Gavrilova, Elena Goiman, Ekaterina Kovalenko, Na’il Saleh, Lyubov Grishina, Vladimir Kozlov

**Affiliations:** 1Research Institute of Fundamental and Clinical Immunology, 14, Yadrintsevskaya St., 630099 Novosibirsk, Russia; 2Department of Clinical Immunology, Novosibirsk State Medical University, 52, Krasny Prospect, 630091 Novosibirsk, Russia; 3Nikolaev Institute of Inorganic Chemistry, 630090 Novosibirsk, Russia; e.a.kovalenko@niic.nsc.ru; 4Department of Chemistry, College of Science, United Arab Emirates University, Al Ain P.O. Box 15551, United Arab Emirates; n.saleh@uaeu.ac.ae

**Keywords:** nanoparticles, cucurbiturils, blood cells, drug delivery, immune system

## Abstract

Cucurbiturils are a family of macrocyclic oligomers capable of forming host–guest complexes with various molecules. Due to noncovalent binding to drug molecules and low toxicity, cucurbiturils has been extensively investigated as potential carriers for drug delivery. However, the immune system’s interactions with different drug carriers, including cucurbiturils, are still under investigation. In this study, we focused on cucurbiturils’ immunosafety and immunomodulation properties in vivo. We measured blood counts and lymphocyte subpopulations in blood, spleen, and bone marrow, and assessed the in vivo toxicity to spleen and bone marrow cells after intraperitoneal administration to BALB/c mice. When assessing the effect of cucurbit[6]uril on blood parameters after three intraperitoneal injections within a week in laboratory animals, a decrease in white blood cells was found in mice after injections of cucurbit[6]util, but the observed decrease in the number of white blood cells was within the normal range. At the same time, cucurbit[7]uril and cucurbit[8]uril did not affect the leukocyte counts of mice after three injections. Changes in the number of platelets, erythrocytes, and monocytes, as well as in several other indicators, such as hematocrit or erythrocyte volumetric dispersion, were not detected. We show that cucurbiturils do not have immunotoxicity in vivo, with the exception of a cytotoxic effect on spleen cells after сucurbit[7]uril administration at a high dosage. We also evaluated the effect of cucurbiturils on cellular and humoral immune responses. We founded that cucurbiturils in high concentrations affect the immune system in vivo, and the action of various cucurbiturils differs in different homologues, which is apparently associated with different interactions in the internal environment of the body.

## 1. Introduction

One of the possible ways to create drug delivery systems is to use nanoscale cavitands capable of host–guest complexation with drugs. Drug complexes can be obtained with various cavitand molecules: cyclodextrins, calixarenes, cucurbiturils, crown ethers, cryptophanes, pillararenes, etc. [1,2,3]. Among cavitands, cucurbiturils have advantages, such as the ability to form strong complexes with various compounds [4,5,6,7]. Cucurbit[n]uril (CB[n], C_6n_H_6n_N_4n_O_2n_, n = 5–10), a class of compounds first discovered by Behrend et al. in 1905, have recently emerged as promising macrocyclic containers formed by the acid-catalyzed condensation of n glycoluril units with formaldehyde [8]. The pumpkin-shaped CB[n]s are caged compounds, the skeleton of which consists of n glycoluril units linked by a pair of methylene groups, providing different outer diameters ranging from 4.5 to 12.5 for CB[5] to CB[10], respectively, and allowing guest molecules of various sizes to be included within the cavity [9,10,11,12,13]. The internal hydrophobic cavity is responsible for complexes with smaller guest molecules through hydrophobic interactions, whereas the two portals (which are lined by urea do carbonyl groups) of pumpkin-shaped CB[n] are responsible for complexation with positively charged molecules through ion–dipole interactions [9,10,14,15].

The average value of binding capacity (the concept is used in characterizing guest–host–guest interaction and receptor–drug, enzyme–substrate interactions) for cucurbit[6]uril(CB[6]) when studying the connection with 56 guest molecules is Ka = 10^3.8±1.5^ M^−1^ [11,16]. A comparison of the complexes of CB[6] and α-cyclodextrin, also capable of “host–guest” complexation, showed that CB[6] forms stronger noncovalent bonds (with the exception of hexanol) with molecules having a positive charge [17,18]. The complexation of CB[6] has a number of characteristics that are similarly applicable to the entire family. Since the portals of the molecule have some negative charge, the formation of associates with positively charged particles occurs in these areas [19]. In the cavity area, the binding of hydrophobic molecules and their particles occurs. The relative mobility and proximity of three binding regions, two for positively charged groups and one for hydrophobic residues, imparts high selectivity when binding to cucurbiturils to form host–guest complexes. When studying the binding constants of a number of alkylammonium ions with CB[6] in a solution of HCO_2_H/H_2_O (1:1), values from 10^1^ to 10^7^ M^−1^ were obtained [16]. Similar features of complex formation will be characteristic of other cucurbiturils due to their similar structure.

Since cucurbit[n]urils are a series of homologues that differ in the number of glycoluril fragments that form this molecule, and hence in the size of the cavity [19], cucurbit[6]uril, cucurbit[7]uril, and cucurbit[8]uril (CB[6], CB[7], and CB[8], respectively) have the most suitable sizes for complexation and are most often used to create drug delivery systems (Appendix A). 

The bioavailability of many drugs is low due to poor solubility. CB[7] has good solubility in aqueous media, while CB[6] and CB[8] are poorly soluble. However, in the presence of a number of ions which are abundant in biological media, the solubility of CB[6] increases significantly [10].

Guest–host complexes of CB[7] and CB[8] with a guest dye molecule can cross the cell membrane [20]. It is likely that other CB[n] guest–host complexes will be similarly capable of passing through the cell membrane, and therefore may be effective carriers for drug transport. Thus, the assessment of the biological safety of cucurbiturils is relevant. It is now known that cucurbiturils have demonstrated low toxicity in a lot of in vitro and in vivo studies [21,22,23,24,25,26]. The use of cucurbiturils at very high doses may cause myotoxicity and neurotoxicity, but there are no signs of toxicity at standard concentrations used in complexation with drugs. 

However, in addition to immunotoxicity, it is also necessary to investigate the possible immunomodulatory properties of delivery systems, since the immunostimulatory or immunosuppressive effects of the system can either enhance or weaken the effect of the delivered drug. Recent studies have shown that the closest analogues of cucurbiturils, cyclodextrins, have anti-inflammatory properties [27]. On the other hand, it is currently known that cyclodextrin can act as an immunostimulant, with prolonged exposures (rats taking 0.4 g/kg/day for three months) causing increased monocyte and overall white blood cells counts [28], and a single co-administration with an antigen (albumin) increasing the proliferation of T helpers 2 (Th2) and T follicular helpers lymphocytes [29]. Investigating the possible immunomodulatory effect of nanosized cavitands is important for the development of new delivery systems and should be considered when assessing indications and contraindications for the treatment of various diseases.

In previous studies, we evaluated the possible immunomodulatory effect of cucurbiturils [30,31,32,33], and it was shown that during cultivation in the presence of CB[n], the proliferative activity of cells increased and the expression of HLA-DR on lymphocytes increased, which indicates an immunostimulatory effect. It was shown that cucurbiturils had practically no immunosuppressive effect under in vitro conditions, except for a slight decrease in HLA-DR expression and the production of reactive oxygen species by T-helpers in stimulated cultures [30,33]. However, for macrocyclic cavitands and for cucurbiturils it has been demonstrated that some properties only manifest themselves under in vivo conditions. Therefore, it seems relevant to study the immunosafety and immunomodulatory properties of cucurbiturils in vivo.

## 2. Materials and Methods

### 2.1. Materials

Cucurbit[n]urils (n = 6, 7, 8; CB[6]·10H2O, MW = 1177; CB[7]·10H2O, MW = 1343; CB[8]·10H2O, MW = 1509) were synthesized at the Nikolaev Institute of Inorganic Chemistry SB RAS (Novosibirsk, Russia) according to a procedure described by A. Day and coauthors [34].

### 2.2. NMR Spectroscopy

Purity control was carried out by ^13^C NMR spectroscopy (250 MHz, 12M HCl, DMSO-internal standard) (ppm):

For CB[6] sample δ = 53.61 (s, 16 CH_2_), 72.28 (s, 16 CH), 158.36 (s, 16 CO);

For CB[7] sample δ = 54.82 (s, 16 CH_2_), 73.26 (s, 16 CH), 158.69 (s, 16 CO);

For CB[8] sample δ = 55.80 (s, 16 CH_2_), 74.04 (s, 16 CH), 159.16 (s, 16 CO).

### 2.3. IR Spectroscopy

The IR spectra are typical of cucurbit[n]uril-containing compounds [35,36]. The strong and broad stretching band in the 3700–2800 cm^−1^ region is attributed to the presence of different types of water molecules involved in hydrogen bonding. Bands corresponding to vibrations in a CB[n] molecule are observed in the 1800–400 cm^−1^ region. The typical vibrational bands in the 1709–1741 cm^−1^ region are assigned to the (C=O) vibration [37].

CB[6]·10H_2_O (KBr, ν/cm^−1^): 3454, 3003, 2897, 1740, 1477, 1416, 1376, 1327, 1295, 1259, 1234, 1190, 1149, 1039, 967, 825, 801, 758, 675, 630, 447.

CB[7]·10H_2_O (KBr, ν/cm^−1^): 3459, 3002, 2923, 2851,1733, 1476, 1419, 1375, 1324, 1293, 1234, 1192, 1153, 1073, 1028, 990, 967, 825, 806, 759, 674, 629, 450.

CB[8]·10H_2_O (KBr, ν/cm^−1^): 3446, 3003, 2923, 1722, 1474, 1425, 1376, 1319, 1293, 1231, 1191, 1156, 1077, 1027, 994, 969, 906, 830, 809, 757, 674, 631, 444.

### 2.4. Elemental Analysis

CB[6]·10H_2_O anal. calc. for C_36_H_56_N_24_O_22_, %: C, 12.0; H, 1.0; N, 14.0. Found: C, 12.2; H, 1.1; N, 14.1%. 

CB[7]·10H_2_O anal. calc. for C_42_H_62_N_28_O_24_, %: C, 12.0; H, 1.0; N, 14.0. Found: C, 12.1; H, 1.0; N, 13.9%. 

CB[8]·10H_2_O anal. calc. for C_48_H_68_N_32_O_26_, %: C, 12.0; H, 1.0; N, 14.0. Found: C, 12.2; H, 1.1; N, 14.1%. 

### 2.5. Mice

BALB/c male mice aged 2–3 months were used in the work. The mice were kept in standard vivarium conditions (free access to food and water and a 12 h day/night cycle). For this study, cucurbiturils were diluted in phosphate-buffered saline and administered intraperitoneally to laboratory animals. It should be noted that CB[6] and CB[8] have poor solubility in water; however, the media and buffer solutions used in assessing the immunomodulatory properties contain various ions, including Na+, that can increase the solubility of cucurbiturils. However, the solubility limit of CB[8] did not allow for obtaining a solution with a concentration higher than 0.2 mM, which made it difficult to comprehensively assess the immunomodulatory properties of this cavitand.

The animals were divided into groups of 5–7 mice which were injected intraperitoneally three times during a week, the first group with 0.25 mL of phosphate-buffered saline (PBS), the second group with 4 M CB[6] solution, the third with the second group of 4 M CB[7] solution, and the fourth with the second group of 4 M solution of CB[8].

Blood sampling was carried out one day after the third injection. The indicators of the cellular composition of the blood were assessed, namely the number of leukocytes, erythrocytes, platelets, granulocytes (neutrophils, eosinophils, basophils), and lymphocytes, as well as the level of hemoglobin and the average concentration of hemoglobin in the erythrocyte and hematocrit. On the next day after the third injection, the animals were sacrificed by decapitation, after which a set of samples (blood, spleen, bone marrow) was taken. The isolation of blood mononuclear cells was performed by centrifugation on a Ficoll–Urografin density gradient. The isolation of bone marrow and spleen cells was carried out using standard techniques. Bone marrow was washed from mouse femurs with phosphate buffer and resuspended with a syringe. To obtain a homogeneous suspension, the cells were broken up by passing the suspension repeatedly through needles of different diameters, and then passing through a cell filter (30 μm). The spleen was crushed with a homogenizer, the resulting suspension was filtered, then the erythrocytes were lysed using a lysis buffer, after which cells were washed twice with phosphate buffer. Next, cells were counted. 

### 2.6. Cytotoxicity Assay

Three million cell suspension was used for LDH, and the study was carried out according to the instructions specified in the kit. LDH-Cytox™ Assay Kit (BioLegend, San Diego, CA, USA) cells (splenocytes, bone marrow cells) at a concentration of 0.5 million/100 µL were plated in a 96-well plate (3 wells for positive control, 3 wells for sample). Next, lysis buffer was added to the wells to evaluate the positive control. After 30 min incubation with lysis buffer and 30 min incubation with assay buffer, the reaction was stopped by adding stop solution and absorbance was measured using an Infinite F50 microplate reader (Tecan, Grödig, Austria) at 490 nm. The average optical density of each set of wells was calculated in triplicate and the value of the background control was subtracted. The percentage of cytotoxicity was calculated using the following equation:Cytotoxicity (%) = ((A − C)/(B − C)) × 100A: test substance; B: positive control; C: negative control.

### 2.7. Flow Cytometry

Samples of blood, spleen, and bone marrow cells were stained with fluorescently labeled monoclonal antibodies. The following anti-mouse antibodies were used: FITC-anti CD45 (clone S18009F), APC-anti-CD3 (clone 17A2), PerCP/Cyanine5.5 anti CD4 (clone GK1.5), PE/Cy7 anti CD16 (clone S17014E) and PE anti-CD19 (clone 1D3/CD19) from Biolegend, USA. Monoclonal antibodies were added to 100 μL of cell suspension containing at least 2 × 10^6^ cells. After 30 min incubation at room temperature in the dark, cells were washed and analyzed. Analyses were performed using a six-color FACSCanto II (Becton Dickinson, Franklin Lakes, NJ, USA) and FACSDiva software 6.1.2 software (Becton Dickinson, USA).

### 2.8. IgM-Plaque-Forming Cell (IgM–PFC) and Delayed-Type Hypersensitivity (DTH) Assay

To assess the possible effect of CB[n] on cellular and humoral immunity in vivo, mice were immunized with thymus-dependent antigen in a 0.25% solution of sheep red blood cells (SRBCs) (ZAO ECOLab, Moscow, Russia) intraperitoneally; immunization was carried out the next day after the second injection and 2 days before the third injection of cucurbiturils. The results were evaluated on the eighth day after the first administration of the drug.

The humoral immune response was assessed at the peak of the response, on the fourth day (IgM-PFC), by the number of plaques (i.e., clear areas of hemolysis around each antibody-forming cell). First, a single cell suspension was prepared from each spleen. Next, an incubation mixture was prepared: 500 µL of cell suspension, 500 µL of complement (ZAO ECOLab, Russia), and 500 µL of SRBC suspension (12 × 10^8^ SRBCs/mL). The components were mixed, and the mixture was poured into glass chambers made of two glass slides glued together along the longitudinal sides using a mixture of wax and paraffin. When filling with a mixture, the volume of the chamber was taken into account. The chambers were incubated at 37 °C for 90 min. The plaques were counted under a binocular loupe according to the formula Absolute PFC = (A × B × C)/D, where A is the number of plaques per chamber, B is the volume of the cell suspension, C is the dilution of the cell suspension, and D is the volume of the chamber.

The cellular immune response was assessed by the severity of the DTH reaction, namely, paw edema after the administration of a second dose of SRBCs to previously sensitized animals. After four days, animals were challenged with 50 µL 50% SRBC suspension in the left hind foot pad. The right foot pad was injected with the same volume of PBS to serve as a control for nonspecific swelling. The footpad thickness was measured with a microcaliper 24 h after the challenge. The ratio of the thickness between the left foot pad and right foot pad was used as a measure of DTH reaction.

### 2.9. Statistical Analysis

All experimental data were expressed as means ± standard error of the mean (SEM) or median with interquartile range. Differences between groups were evaluated for statistical significance using a Student’s *t*-test or Mann–Whitney test when the data were not normally distributed. A *p*-value < 0.05 was regarded as the minimum criteria for statistical significance.

## 3. Results

### 3.1. Immunosafety Findings

First, we evaluated the effect of cucurbit[n]urils administered parenterally on the blood parameters of laboratory animals of the BALB/c line. When assessing the effect of cucurbit[7]uril on blood parameters after three intraperitoneal injections within a week in laboratory animals, a decrease in white blood cells was found in mice after injections of CB[6] from 7013 ± 1437 to 3400 ± 461 cells/μL (Figure 1a,b). The typical leukocytes count in mice is 2000 to 10,000 per μL [38]. Therefore, the observed decrease in the number of white blood cells is within the normal range (Figure 1a). CB[6] did not cause a statistically significant decrease in the relative number of leukocyte subpopulations such as lymphocytes, monocytes and neutrophils (Figure 1b). However, trends were observed for a decrease in the number of neutrophils after the third injection in animals that were injected with CB[6]. CB[7] and CB[8] did not affect the leukocyte counts of mice after three injections. Changes in the number of platelets, erythrocytes, and monocytes, as well as in several other indicators, such as the average amount of hemoglobin per red blood cell, were not detected (Appendix A).

The assessment of the immunotoxic effect of cucurbiturils on the central and peripheral organs of the immune system was carried out using the method of determining the activity of LDH. It was found that three times the intraperitoneal administration of cucurbiturils at high concentrations did not have a toxic effect on mouse bone marrow cells (Figure 2a). However, it has been shown that the administration of CB[7] can lead to toxic damage to spleen cells, since a mild toxic effect on splenocytes was found (Figure 2b). However, CB[6] and CB[8] did not have a toxic effect on spleen cells. At the same time, the total level of LDH in the serum of mice was increased after the administration of CB[6], CB[7] and CB[8] (Appendix A), which indicates a possible toxic effect of high doses of cucurbituril on other organs and tissues. It is known that cucurbiturils have myotoxicity and neurotoxicity [24], so the increase in LDH in the blood serum is more likely to be associated with a toxic effect on muscles and the nervous system, rather than the immune system.

### 3.2. Lymphocytes

The impact of cucurbiturils on the subpopulation composition of peripheral blood, spleen, and bone marrow lymphocytes of laboratory animals was assessed. It was found that after the injections of CB[6] in the peripheral blood there was a decrease in CD4^+^ T-helpers, and at the same time an increase in the proportion of CD19^+^ B-lymphocytes compared with the control (Figure 3). CB[7] and CB[8] did not affect the subpopulation composition of peripheral blood lymphocytes. 

Since the peripheral blood contains only a small proportion of the body’s lymphocytes, the next step was to assess the subpopulation composition of lymphocytes in the spleen, where lymphocytes are found in large numbers in the white pulp. In the case of assessing the influence of the subpopulation composition of spleen cells, CB[6] and CB[7] increased the number of B-lymphocytes, and CB[8] increased the amount of B-cells and cytotoxic T-lymphocytes (Figure 4).

It is known that the formation of new naive B lymphocytes occurs in the bone marrow. In bone marrow, CB[6] and CB[7] led to an increase in the relative number of B-lymphocytes, while the number of CD16^+^ NK cells decreased proportionally (Figure 5). CB[8] also reduces the relative number of NK cells.

Thus, our studies have shown that CB[6] and CB[7] can increase the percentage of B cells in the organs of the immune system, both in the spleen and in the bone marrow. The treatment with CB[8] leads to an increase in the relative number of B cells and cytotoxic T-lymphocytes in mouse spleen. 

### 3.3. Immune Responses Findings

Next, we explored the effect of cucurbiturils on different types of immune responses after SRBCs antigen administration in mice (Figure 6). Treatment with CB[6] reduced the level of PFC in the spleen of laboratory animals compared to control. While CB[7] and CB[8] did not affect the amount of PFCs, the observed slight decrease was not significant. On the contrary, the severity of the DTH reaction increased with the administration of cucurbiturils, while significant changes were observed with the administration of CB[7] and CB[8]. 

Thus, CB[6] can suppress the severity of the humoral immune response to the antigen. However, the treatment with CB[7] and CB[8] does not suppress the immune response, but stimulates the severity of the DTH reaction.

## 4. Discussion

The present study was conducted to explore the immunotoxicity and immunomodulation properties of CB[n] in vivo. The intraperitoneal administration of cucurbiturils to mice did not lead to changes in blood parameters, and did not cause the death of immune cells, with the exception of spleen cells when treated with CB[7]. It should be noted that high concentrations of cucurbiturils were used in this study; when used for drug delivery, these high concentrations are unlikely to be required. 

In previous in vitro studies, cucurbiturils did not show immunotoxicity [30,31]. In the present animal study, the low immunotoxicity of cucurbiturils was confirmed. Therefore, cucurbiturils have practically no immunotoxicity, except for the cytotoxic effect on spleen cells in vivo when CB[7] is administered at a high dose, and the trend towards an increase in the number of apoptotic cells in cultures cultivated in the presence of CB[8] [31]. The results may indicate the immunosafety of using cucurbiturils for drug delivery.

We also evaluated the influence of the immunomodulatory action of cucurbiturils. It was demonstrated that different homologues have different effects on the immune system. Based on the obtained data in vitro, CB[8] causes a decrease in the level of IFN-gamma [32], which indicates its possible immunosuppressive effect. Since CB[8] is able to suppress cytokine production, a possible immunosuppressive effect requires further study. If immunosuppressive activity is confirmed, CB[8] can be used as a basis for delivery systems for various immunosuppressive drugs. However, when assessing the cellular and humoral immune response in animals after the IP administration of CB[8], no immunosuppression was observed.

In the case of CB[6], its stimulating effect on the B cells and humoral immune response was demonstrated, since it increased the expression of HLA-DR molecules on B-lymphocytes [21], increased the level of spontaneous production of IL-4 by 1.5 times [32], and also increased in vivo the relative number of B cells in the organs of the immune system. At the same time, the severity of the humoral immune response to a certain antigen after the introduction of CB[6] was reduced, which requires studying the mechanisms of this process. Perhaps the strong nonspecific stimulation of B-lymphocytes did not allow the cells to respond to antigen stimulation. 

CB[7] showed mild immunostimulatory properties, enhancing the DTH response in vivo, increasing the relative number of B cells in mice and suppressing the production of IL-10 in PBMC-stimulated culture in vitro [32]. These properties make CB[7] an excellent basis for drug delivery systems for the treatment of infectious and tumor diseases. The complexation of drugs with CB[7] enhanced their antitumor activities [39]. It is known that antitumor drugs can have a direct effect on tumor cells, leading to their death. However, in recent years, due to the spread of immunotherapy, more and more information is emerging about the importance of the role of the immune system in protecting against cancer cells, including during chemotherapy. Extensive evidence suggests that the clinical success of chemotherapy is due not only to toxicity to tumor cells, but also to the restoration of immunosurveillance, which has been sorely neglected in previous preclinical and clinical studies [40,41,42]. Immune cells are an important component of the tumor microenvironment, promoting immunosuppression in relation to tumor structures, and also influencing the effectiveness of therapy [43]. Delivery systems and their components can reduce the number of immune suppressor cells (Tregs, myeloid suppressors, etc.) in the tumor microenvironment, influence the expression of checkpoint inhibitors in tumor tissue, enhance the antigen uptake and maturation of dendritic cells, and also stimulate immune cells involved in antitumor immune responses. In addition, the supramolecular delivery system can enhance the antitumor effect of the drug through the immune system through tumor immunogenic cell death induction [44]. Effects on the tumor microenvironment and induction of immunogenic cell death have been shown for antitumor drug delivery systems based on macrocyclic components, namely cyclodextrin and pillararene [45,46]. In the case of cucurbituril-based systems, it has been demonstrated that CB[7] does not decrease the immunomodulating properties of platinum drugs in vitro [47]. The CB[7]-carboplatin mixture in PBMC-stimulated and non-stimulated cultures significantly reduced the amount and expression of CTLA-4 of FoxP3^+^ regulatory T cells compared to the control [48]. It has also been shown that the complex of CB[7] with tuftsin, a peptide with immunostimulating and antitumor effects, increases the production of IFN gamma, a cytokine involved in the antitumor Th1 immune response [49]. 

Therefore, when developing new drugs for chemotherapy, it is necessary to evaluate the restoration of the antitumor response and immunological surveillance of the tumor, which will allow for a comprehensive assessment of the antitumor effect of the studied constructs even at the stage of preclinical studies. We assume that the use of CB[7] as a drug delivery system can enhance the antitumor effect due to its effect on immune cells in tumor tissue. 

Thus, at high concentrations, cucurbiturils may have an effect on the immune system in vivo. At the same time, the action of various cucurbiturils differs in different homologues, which, apparently, is associated with different interactions with components in the internal environment of the body. Since homologues differ in their effects on the immune system in vivo, further research into the mechanisms of such action is required. It is believed that broader and deeper explorations of the possible immune properties of cucurbiturils could facilitate the further development of new drugs and novel treatment strategies.

## Figures and Tables

**Figure 1 pharmaceutics-16-00127-f001:**
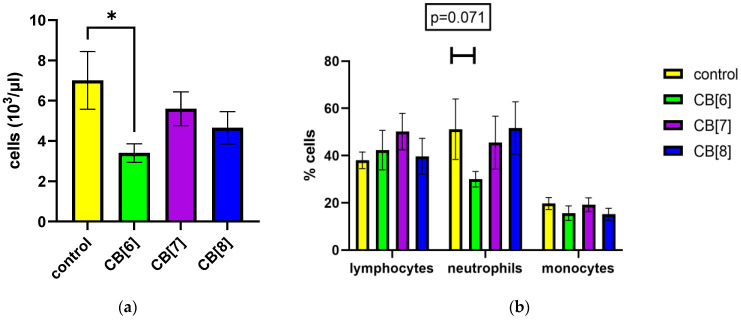
Effect of intraperitoneal administration of cucurbiturils on white blood cell count. (**a**) The number of white blood cells; (**b**) lymphocyte, neutrophil, and monocyte measurements. Data are presented as the median with interquartile range. * Indicates a significant difference (*p* < 0.05) vs. the control.

**Figure 2 pharmaceutics-16-00127-f002:**
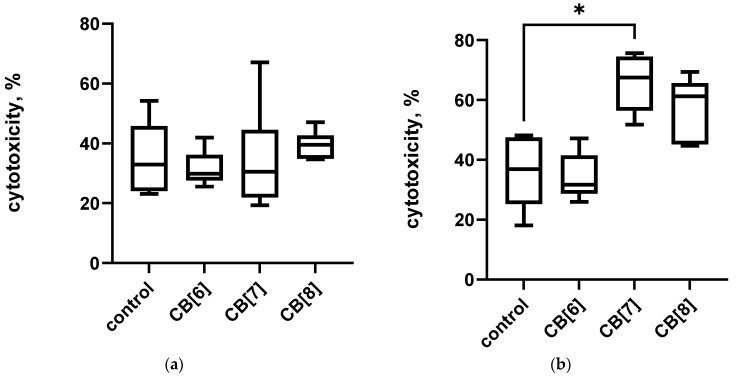
Lactate dehydrogenase (LDH) leakage in cells after intraperitoneal cucurbituril administration. (**a**) Bone marrow; (**b**) spleen. Data are presented as box-and-whisker plots, with boxes extending from the 25th to the 75th percentile, with a horizontal line at the median, while the whiskers extend to the lowest and highest data points. * Indicates a significant difference (*p* < 0.05) vs. the control.

**Figure 3 pharmaceutics-16-00127-f003:**
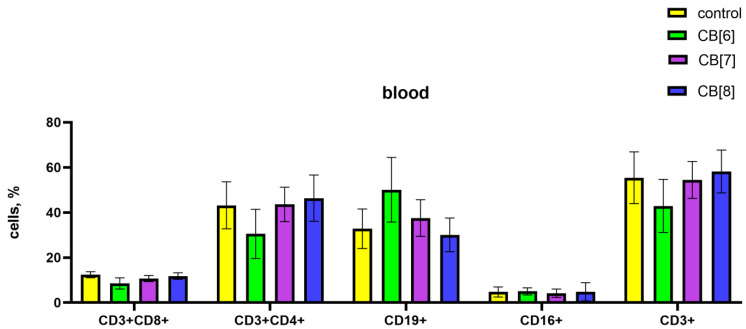
Evaluation of peripheral blood subset percentages in response to the intraperitoneal administration of cucurbiturils. Data are presented as the median with interquartile range.

**Figure 4 pharmaceutics-16-00127-f004:**
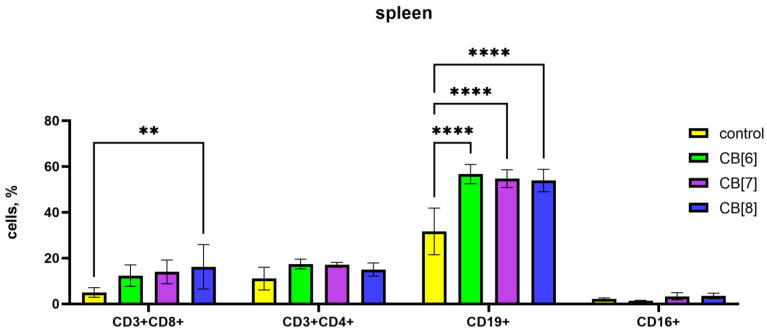
Evaluation of spleen subsets percentage in response to the intraperitoneal administration of cucurbiturils. Data are presented as the median with interquartile range. ** Indicates a significant difference (*p* < 0.01) vs. the control. **** Indicates a significant difference (*p* < 0.0001) vs. the control.

**Figure 5 pharmaceutics-16-00127-f005:**
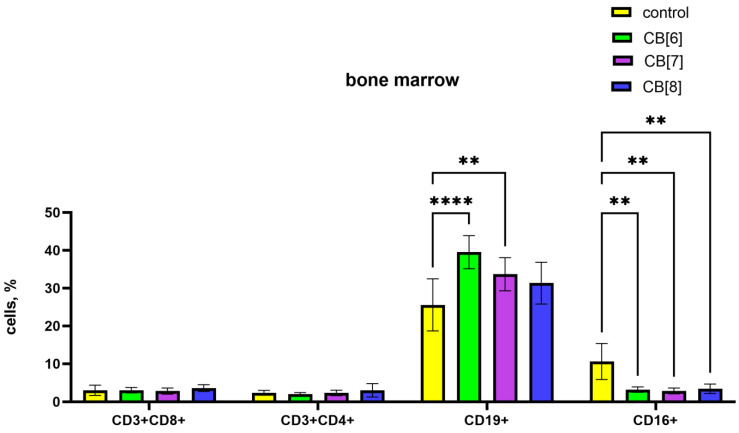
Evaluation of bone marrow lymphocyte subset percentage in response to the intraperitoneal administration of cucurbiturils. Data are presented as the median with interquartile range. ** Indicates a significant difference (*p* < 0.01) vs. the control. **** Indicates a significant difference (*p* < 0.0001) vs. the control.

**Figure 6 pharmaceutics-16-00127-f006:**
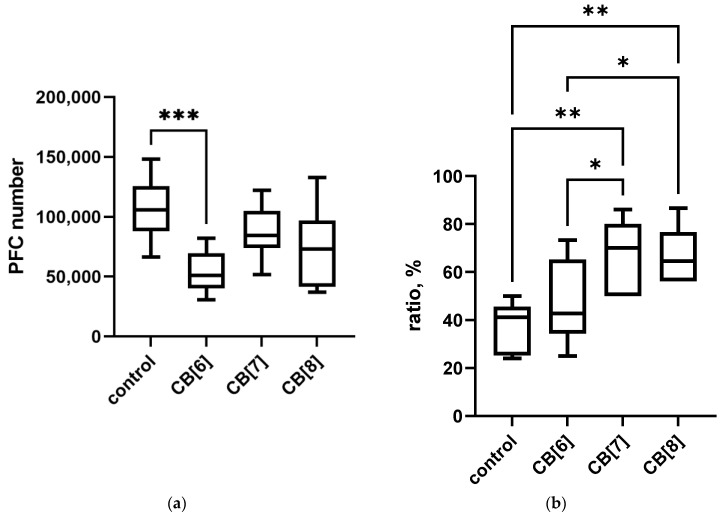
Evaluation of the immune response after intraperitoneal cucurbituril administration. (**a**) IGM-PFC; (**b**) DTH. Data are presented as box-and-whisker plots, with boxes extending from the 25th to the 75th percentile, with a horizontal line at the median, while the whiskers extend to the lowest and highest data points. * Indicates a significant difference (*p* < 0.05) vs. the control. ** Indicates a significant difference (*p* < 0.01) vs. the control. *** Indicates a significant difference (*p* < 0.001) vs. the control.

## Data Availability

The data presented in this study are available in this article (and Appendix A).

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
