# Peer review of "Evaluation of the Immunosafety of Cucurbit[n]uril In Vivo"

_pharmaceutics, 2024, doi:10.3390/pharmaceutics16010127_

Round 1

Reviewer 1 Report

Comments and Suggestions for Authors

The paper authored by Pashkina and colleagues conducts a thorough examination of the immunosafety of cucurbit[n]uril through in vivo experiments. The authors utilize a variety of biological assays, investigating the effects of intraperitoneal administration of cucurbiturils on white blood cells, their impact on the subpopulation composition of peripheral blood, and the evaluation of spleen subsets percentage in response to intraperitoneal administration, among others. This approach showcases commendable organization and potential significance in its field. However, several critical considerations should be taken into account before considering this work for publication:

1. Physicochemical Characterization: The absence of physicochemical characterization for the obtained cucurbit[n]urils is a notable gap. Including such characterization is essential for confirming their synthesis and ensuring a comprehensive understanding of the materials under investigation.

2. Figure 1b Quality: The quality of Figure 1b is suboptimal. A revision and enhancement of this figure are necessary to ensure clarity and improve the visual impact of the results presented.

3. Statistical Analysis for Figure 2: Similar to the other figures, results from Figure 2 should be accompanied by a statistical analysis, providing a more robust interpretation of the data and enhancing the overall reliability of the findings.

4. Exploration of Pharmaceutical Applications: The manuscript could benefit from additional experiments demonstrating the potential pharmaceutical applications of the studied materials, such as drug delivery profiles. In the current scientific landscape, manuscripts that delve into specific applications tend to have a more substantial impact, especially for journals with high impact factors.

By addressing these points, the paper's scientific rigor will be significantly strengthened, ensuring a more comprehensive and impactful contribution to the scientific literature.

Author Response

  1. Physicochemical Characterization: The absence of physicochemical characterization for the obtained cucurbit[n]urils is a notable gap. Including such characterization is essential for confirming their synthesis and ensuring a comprehensive understanding of the materials under investigation.

We have added Physicochemical Characterisation in the manuscript. 

2. Figure 1b Quality: The quality of Figure 1b is suboptimal. A revision and enhancement of this figure are necessary to ensure clarity and improve the visual impact of the results presented

Thanks for the comments, we have made corrections. The figure 1b has been changed.

3. Statistical Analysis for Figure 2: Similar to the other figures, results from Figure 2 should be accompanied by a statistical analysis, providing a more robust interpretation of the data and enhancing the overall reliability of the findings.

We have made the suggested corrections

4. Exploration of Pharmaceutical Applications: The manuscript could benefit from additional experiments demonstrating the potential pharmaceutical applications of the studied materials, such as drug delivery profiles. In the current scientific landscape, manuscripts that delve into specific applications tend to have a more substantial impact, especially for journals with high impact factors.

In this article, we have focused on the immunomodulatory properties of the delivery system itself. We have added additional information on the immunomodulatory properties of host-guest complexes  of macromolecules with drugs and anticancer drug-loaded delivery systems in the discussion section.

Reviewer 2 Report

Comments and Suggestions for Authors

In this manuscript, Pashkina A. et al. examined the in vivo immunosafety and immunomodulation properties of cucurbiturils polymers (cucurbit[6]uril, cucurbit[7]uril and cucurbit[8]uril ), which are drug carrier macromolecules, by using BALB/c mice. They show some different effects on the number of white blood cell. Such immunosafety experiments is important, but several points need to be modified.

There are three “Figure 1” in this manuscript (p5 and p9). In addition, they did not mention about Figure 1a or 1b in line194. So, I was very confused to understand this manuscript. So, please check the correspondence between the figure and the text again.

Author Response

In this manuscript, Pashkina A. et al. examined the in vivo immunosafety and immunomodulation properties of cucurbiturils polymers (cucurbit[6]uril, cucurbit[7]uril and cucurbit[8]uril ), which are drug carrier macromolecules, by using BALB/c mice. They show some different effects on the number of white blood cell. Such immunosafety experiments is important, but several points need to be modified.

There are three “Figure 1” in this manuscript (p5 and p9). In addition, they did not mention about Figure 1a or 1b in line194. So, I was very confused to understand this manuscript. So, please check the correspondence between the figure and the text again.

Thank you for your comments. We apologize for the mistakes, we have made the suggestet corrections.

Round 2

Reviewer 1 Report

Comments and Suggestions for Authors

In my opinion, the manuscript can be published in the present form.